# Colitis-Associated Dysplasia in Inflammatory Bowel Disease: Features and Endoscopic Management

**DOI:** 10.3390/cancers17050784

**Published:** 2025-02-25

**Authors:** Sara C. Schiavone, Livia Biancone, Mariasofia Fiorillo, Andrea Divizia, Roberto Mancone, Benedetto Neri

**Affiliations:** 1Gastroenterological Unit, Department of Systems Medicine, University “Tor Vergata” of Rome, 00133 Rome, Italy; saraschiavone27@gmail.com (S.C.S.); fiorillo93@gmail.com (M.F.); roberto.mancone@yahoo.it (R.M.); benedettoneri@gmail.com (B.N.); 2Department of Surgery, University “Tor Vergata” of Rome, 00133 Rome, Italy; andreadivizia@live.it

**Keywords:** IBD, colitis-associated dysplasia, chromoendoscopy, colorectal cancer, endoscopic surveillance, endoscopic submucosal dissection, endoscopic mucosal resection

## Abstract

A higher risk of colitis-associated dysplasia is observed in patients with long-standing colonic inflammatory bowel disease (IBD). The “inflammation-dysplasia-carcinoma” sequence reported in IBD accounts for the peculiar characteristics of colitis-associated colorectal lesions. Specific endoscopic surveillance strategies tailored to patients’ individual neoplastic risk, including dye and virtual chromoendoscopy, have been proposed and adopted to achieve early diagnosis. In cases of early diagnosis, IBD-associated dysplastic lesions can be successfully treated through endoscopic resection, even though this procedure is more challenging than in the general population. The resection of visible dysplastic lesions in one fragment (en bloc) is associated with a lower risk of local recurrence. Therefore, particularly for large (>20 mm) lesions, endoscopic submucosal dissection, characterized by higher rates of en bloc resection, is the preferred technique, although challenging. Future technical and technological advances will allow optimization of both endoscopic surveillance and resective strategies, leading to earlier diagnosis and more effective treatments of colitis-associated dysplasia.

## 1. Introduction

Inflammatory bowel disease (IBD), including ulcerative colitis (UC) and Crohn’s disease (CD), is characterized by a relapsing course and chronic inflammation of the gastrointestinal tract [1]. IBD-related colitis with chronic inflammation involving at least one third of the colon plays a key role in the higher risk of dysplasia and colorectal cancer (CRC) in these patients [2]. CRC risk in IBD colitis is proportional to the extent of colonic involvement, disease duration and severity of inflammation (both endoscopic and histological) [3,4]. Additional main risk factors for CRC, including those common to the general non-IBD population (age, familial history of CRC, personal history of adenoma, dysplasia or CRC) or which are colitis-associated (primary sclerosing cholangitis, PSC or colonic strictures), have also been reported [5].

## 2. Epidemiology

A meta-analysis from 2001 by Eaden et al. [6] reported a cumulative 2%, 8% and 18% risk of CRC in UC patients after 10, 20 and 30 years of disease, respectively. However, the risk of developing CRC seems to be lower in CD colitis than in UC. In these patients, in a meta-analysis by Canavan et al. [7], a cumulative risk of 2.9%, 5.6% and 8.3% at 10, 20 and 30 years was reported, respectfully. However, more recent studies suggest a lower cumulative risk of CRC in IBD patients with a 30-year history of disease, being reported as 7% in UC and 2% in CD [8]. This observation may be related to the marked improvement in both treatments and endoscopic surveillance [9,10]. Indeed, several studies emphasized the correlation between decreased inflammatory activity, induced by medical treatments, and a reduced risk of developing CRC [11,12,13]. Despite this, CRC still represents the most frequent indication for colectomy and the leading cause of mortality in these patients [14], contributing to about 15% of IBD-related mortality [15].

## 3. CRC Development Pathways in IBD

Different from sporadic CRC, which develops through an “adenoma-carcinoma” sequence, IBD colitis-associated carcinogenesis currently appears to be mainly related to an “inflammation-dysplasia-carcinoma” sequence. Indeed, chronic inflammation may determine oxidative, stress-induced damage to DNA, resulting in activating tumor-promoting genes and inactivating tumor suppressor genes [2]. Unlike sporadic CRC, where dysplastic lesions usually arise in one or two focal areas of the colon, dysplasia or cancer in IBD are frequently diffuse or multifocal due to a process known as the “field effect” [16].

However, the main molecular pathways involved in the development of sporadic CRC, namely chromosomal instability (CIN), microsatellite instability (MSI) and CpG island methylator phenotype (CIMP), also contribute to the development of colitis-associated colorectal cancer (CAC) [2]. Emerging evidence suggests that CIN and MSI, the two major pathways involved in the development of sporadic CRC, are also involved in CAC with approximately the same frequency (85% CIN and 15% MSI) [17]. CIN results in abnormal segregation of chromosomes and an abnormal DNA content (aneuploidy), which often cause loss of chromosomal material (loss of heterozygosity) and contribute in turn to the loss of function of key tumor suppressor genes, such as APC and p53. These genes may also become nonfunctional due to mutations [2].

What appears to be different is the timing and frequency of gene alterations more frequently observed in sporadic and colitis-associated CRC. While loss of APC function is typically an early event in sporadic CRC, gene mutation occurs less often and later in the dysplasia-carcinoma sequence of the CAC. The loss of p53 gene function has been reported to occur later in sporadic CRC, and it is believed to be the defining event which drives the adenoma-to-carcinoma sequence. This mutation has been shown to frequently occur earlier in the development of CAC, often being detected in non-dysplastic colonic mucosa or in areas defined as indefinite for dysplasia [3]. The MSI pathway involves the primary loss of function of genes which usually repair DNA base-pair mismatches, which occurs in dividing cells during the normal process of DNA replication [2]. Epigenetic alterations can also contribute to altered gene expression in colon carcinogenesis. Indeed, hypermethylation of CpG islands may determine silencing and, therefore, loss of gene expression [2]. Different genes controlling the cell cycle, cell adhesion and DNA repair are known to be methylated in sporadic CRC [18]. In CAC, methylation of CpG islands in several genes seems to precede dysplasia and is more widespread throughout the mucosa of UC patients [19].

## 4. Dysplasia

In the gastrointestinal tract, dysplasia is defined histologically as the replacement of the intestinal epithelium with neoplastic but noninvasive epithelium [20]. In IBD, dysplasia has been classified into five categories: negative for dysplasia, indefinite for dysplasia, low-grade dysplasia (LGD), high-grade dysplasia (HGD) or invasive cancer [20]. The difference between low- and high-grade dysplasia relies on the distribution of the nuclei within the cells. Low-grade dysplasia is defined by the nuclei localized in the basal half of the cells, while in high-grade dysplasia nuclei, they are mainly in the basal and upper half [20]. Macroscopic and microscopic chronic colonic inflammation are currently considered among the most relevant risk factors for LGD, which can hesitate in HGD and progress to CRC [21]. This sequence of events is reputed to be faster in CAC when compared with sporadic CRC [21]. The sequence IBD-related “inflammation-dysplasia-cancer” sequence has been extensively investigated [5]. The risk of progression from colonic dysplasia to cancer is reported to be low, but this risk increases over time in patients with IBD. Studies have shown that that the risk of CRC in IBD increases by 0.5–1.0% yearly and significantly after 8–10 years of disease duration [6]. Overall, dysplasia is a well-established histological stage in the development of CRC, and its detection during colonoscopy should prompt appropriate management to prevent progression to CAC. Therefore, appropriate and timely surveillance by using high-quality endoscopic and histological examinations is a crucial step in order to prevent the development of CAC. In order to optimize this goal, these examinations should be tailored to a patient’s basis and performed by experienced, dedicated physicians.

## 5. Surveillance

Historically, the traditional technique used to perform surveillance consisted of standard-definition white light endoscopy (SD-WLE) with multiple random biopsies (random 4-quadrant biopsies every 10 cm from the caecum to the rectum, for a minimum of 32), plus targeted biopsies of visible lesions [22]. Thereafter, dye chromoendoscopy (DCE) with target biopsies has been recommended over SD-WLE for detecting dysplasia through SCENIC consensus [23]. In DCE, a contrast agent (indigo carmine 0.03–0.1% or methylene blue 0.04–0.1%) is applied on the colonic mucosa, allowing detection of the less visible polypoid and non-polypoid lesions [23,24].

More recently, virtual chromoendoscopy systems (VCEs), including iSCAN, NBI and BLI/LCI, have been developed. VCEs allow instant digital staining, therefore enhancing the glandular and vascular pattern of the colonic epithelium [25,26]. This technique is less time-consuming and less expensive than DCE [26,27]. Different from the SCENIC consensus, the European Society of Gastrointestinal Endoscopy (ESGE) equally recommends the use of DCE or VCE for optimal surveillance. Sampling of any suspected lesion in IBD is also suggested [28]. These recommendations derive from trials showing the superiority of DCE over SD-WLE for the detection of neoplasia [29] and from studies reporting no clear superiority between DCE and VCE [30,31]. To date, SD-WLE is considered to be inadequate for CRC surveillance in IBD, while also considering the introduction of high-definition white light endoscopy (HD-WLE). HD-WLE is characterized by a wider field of vision, a higher pixel density and faster line scanning, therefore producing sharper images and less artifacts [25,32]. These features of HD-WLE are associated with improvement in the detection of dysplastic lesions, particularly when compared with SD-WLE [33]. Moreover, current data suggest that HD-WLE may allow for equivalent detection of dysplasia, even when compared to DCE or VCE [25,34]. Indeed, a prospective randomized controlled trial (RCT) comparing HD-WLE with HD-DCE and VCE with I-SCAN for detecting neoplastic lesions during IBD surveillance colonoscopy reported that the HD-WLE neoplasia detection rate was not inferior to that of either DCE or VCE [30].

In a different prospective RCT, HD-DCE with indigo carmine and HD-WLE alone were compared in terms of ability to detect neoplastic lesions during IBD surveillance colonoscopy with random biopsies. In this RCT, the superiority of HD-DCE versus HD-WLE for detecting dysplasia per colonoscopy was reported [35]. The benefits of VCE over new-generation HD-WLE have not yet been assessed in RCTs.

Results from more recent meta-analyses support the role of DCE. A network meta-analysis by Sinopoulou et al. [36] suggested that HD-DCE is the only modality for IBD surveillance, with suggested evidence (but a low certainty) of a potential higher ability to detect more dysplastic lesions than HD-WLE. An updated meta-analysis of RCTs comparing dysplasia detection using DCE versus HD-WLE in IBD patients undergoing surveillance colonoscopy also indicated HD-DCE to be a superior strategy for overall dysplasia detection in IBD, even with HD scopes [37].

All these techniques have increased the detection of precancerous colorectal lesions. However, target sampling of any suspicious lesion is still recommended when performing surveillance in IBD. However, random sampling of the colon and rectum is no longer required. Indeed, data from two RCTs and a retrospective cohort study reported the same dysplasia detection rate in patients undergoing VCE when comparing random plus targeted biopsies versus target biopsies alone [38,39,40]. The diagnostic yield of the dysplasia detection rate when using random biopsies was reported to be as low as 0.2% [41]. In extremely high-risk patients (i.e., previous dysplasia, concomitant PSC, active inflammation and scarred colon), quadrantic biopsies every 10 cm are still indicated [5]. In this particular sub-population, the addition of random biopsies is associated with an increased diagnostic yield of colorectal dysplasia [41]. Indeed, random biopsies in addition to VCE or DCE should be reserved only for special situations, such as surveillance after endoscopic resection of large (>2 cm) non-polypoid lesions [5].

Appropriate surveillance colonoscopy needs to be performed during quiescent IBD activity after adequate bowel preparation [28], even when obtained with very low-volume (VLV) polyethylene glycol (PEG)-based bowel preparations [42]. Uncontrolled microscopic inflammation and retained stools represent two relevant patient-related variables responsible for suboptimal mucosal assessment. In this situation, in appropriately trained hands, non-targeted four-quadrant biopsies can be avoided [28]. An additional key point for optimizing visualization of the colonic mucosa is an adequate colonoscope withdrawal time, even if no specific recommendations have been proposed regarding the optimal time in this setting of patients. The standard 6 min retraction time could not be adequate, since a DCE-examination also includes time for spraying, suctioning of the dye and careful inspection of the mucosa, thus adding an average time of 11 min [43]. Therefore, an expert consensus in 2019 suggested as a good practice a total withdrawal time of at least 17 min for this technique [26]. Additional upsides when using VCE include the lower amount of time needed for the procedure (about 7 min less than DCE) [31] and better surveillance in cases of poorer bowel preparation.

Finally, preliminary recent findings suggest that artificial intelligence software, originally developed for detecting sporadic neoplastic colonic lesions, may be promising for detecting IBD-associated dysplasia. Successful detection of colonic dysplasia has currently been described in case reports including UC patients regarding the application of EndoBRAIN, a CADe system, in endoscopy and endocytoscopy [44,45].

## 6. Clinical Features and Risk Factors

CAC differs from sporadic CRC for several features. CAC does indeed more frequently occur at a younger age [10,46,47], is more often synchronous [40] and shows a higher frequency of mucinous or signet ring cell histology [2,48,49]. These histological features of CAC are more frequently associated with an aggressive course. In addition, rather than developing from a polypoid lesion, CAC often arises from flat dysplastic lesions with undefined margins [50].

These observations account for the need for appropriate CAC surveillance by using optimized endoscopic procedures with biopsy sampling in experienced hands. Appropriate timing of endoscopic surveillance, modulated according to specific risk factors in each patient, is also required for early detection of dysplastic lesions in patients with colonic IBD.

As CAC is rarely encountered within the first 8 years of disease onset [51,52,53,54], current European Crohn’s and Colitis Organization (ECCO) guidelines recommend the first screening colonoscopy 8 years after IBD onset in order to assess the disease extent and exclude dysplasia [5]. After the initial screening, the timing for subsequent colonoscopies needs to be defined in each patient according to specific risk factors [5].

Family history of CRC is a known risk factor for advanced colorectal neoplasia (aCRN), which includes both HGD and CRC in IBD. This evidence was derived from multivariate analyses of case–control and cohort studies [55]. This risk is relevant in both the first and second degrees, although the higher risk is for patients with a first-degree relative with CRC occurring before 50–55 years of age [56,57].

Among the disease-related factors, extensive colonic disease (>50% colonic involvement in CD and pancolitis in UC) is associated with a 2–3-fold higher risk of IBD-associated CRC when compared with lesser-extent CD colitis and left-sided UC. However, both carry a higher aCRN risk with regard to CD or UC limited to the rectum [55].

IBD patients with versus without concomitant PSC have an estimated 3–5-fold higher risk of aCRN [55,58,59]. In these patients, the pathogenic mechanisms leading to this increased risk have not been clarified. Changes in bile acid metabolism, intestinal or biliary microbiome or the involvement of the host immune response in the development of both colorectal and biliary tract cancer has been hypothesized in these patients. Compared with patients without PSC, CRC is right-sided more often, and the risk of LGD progression to cancer is higher in patients with PSC [59].

Colonic strictures, particularly long-standing ones, represent an additional risk factor for CRC [5,60]. This is also related to technical difficulties in terms of appropriate biopsy sampling at this level. Therefore, the detection of dysplasia and even CRC within a colonic stricture may often be delayed, even in IBD patients undergoing regular follow-ups. Due to these observations, colonic resection must be considered in IBD patients, particularly UC patients, for developing the colonic structure [60]. This option needs to be extensively discussed with the patient, and the final decision should be tailored on the basis of additional risk factors for CRC for each patient (age, familial CRC, personal history of adenoma, dysplasia or CRC, IBD duration, extent severity of colonic inflammation and stenosis or strictures and PSC) [60].

Post-inflammatory polyps (“pseudopolyps”) have also been previously associated with an increased risk of CAC. Recent observations do not support a relevant role of these lesions as independent risk factors for CRC, and therefore their presence alone has not necessitated the need for heightened surveillance [55,61,62]. However, the role of pseudopolyps in IBD colons as independent risk factors for CRC is still under investigation. A recent monocentric study showed that one fourth of pseudopolyp-like lesions detected during surveillance colonoscopy in patients with longstanding IBD beared dysplastic foci, suggesting the need for further research in this regard [63].

Histologically active inflammation is a major risk factor for CAC [3,64]. Evidence from established surveillance cohorts consistently supports that cumulative long-term severe inflammation (but not active inflammation) at the time of a colonoscopy is a strong independent predictor of CAC [64,65]. Due to the higher cancer risk in colonic areas with current or previous inflammation, IBD patients have a high risk of synchronous and metachronous neoplasia. Once pathologically confirmed dysplasia is detected, the patient should be considered to be at high risk.

For these reasons, the timing of endoscopic surveillance in IBD patients, tailored according to the specific risk for each patient, currently involves the following time intervals. Patients with high-risk features (family history of CRC in a first-degree relative ≤50 years of age, colonic stricture or dysplasia, PSC or extensive colitis with severe active inflammation) should undergo a surveillance colonoscopy once a year [5]. However, patients with intermediate risk factors (extensive colitis with mild-to-moderate endoscopic or histological inflammation or CRC in a first-degree family member >50 years old) should undergo a subsequent surveillance colonoscopy scheduled 2–3 years later [5]. Finally, patients with neither intermediate nor high-risk features (extensive colitis with minimal endoscopic or histological inflammation or colitis affecting <50% of the colon) should have their next surveillance colonoscopy scheduled 5 years later [5].

## 7. Endoscopic Assessment

The endoscopic morphology of dysplasia detected during a surveillance colonoscopy should be defined as “visible” (polypoid or non-polypoid) or “invisible” (i.e., dysplasia found during histological examination in the absence of a visible lesion during the colonoscopy).

In the recent past, raised or polypoid dysplasia in the setting of IBD has been termed “dysplasia-associated lesion or mass” (DALM), which was believed to be associated with a high risk of CRC development [66,67]. Therefore, a diagnosis of DALM usually meant total colectomy for cancer prevention [67]. However, a differential diagnosis between a DALM lesion and sporadic adenoma (i.e., a different polypoid precancerous lesion whose development is unrelated to the underlying IBD colitis) can be challenging even for experienced endoscopists and pathologists. In IBD, the incidence of sporadic adenoma increases with age, as reported in the general population, in which a longer time interval from the occurrence of sporadic adenoma to CRC has been suggested [68]. As for the general population, the endoscopic complete removal of a sporadic adenoma is considered to be an adequate treatment method for preventing the evolution to CRC in IBD [68]. Indeed, recent studies have shown that the risk of CRC is not increased in patients with colitis-associated lesions removed endoscopically. Therefore, in these cases, total colectomy is considered to be unnecessary [69,70]. Taking into account these data, since 2015, in accordance with the SCENIC’s recommendations, the terms “DALM”, “adenoma-like” and “nonadenoma-like” should be abandoned [23].

According to the recent ECCO guidelines, identified visible dysplastic lesions should be morphologically characterized using the “Five S” method: site, size, shape, surface and surroundings [5]. This is used to establish whether a lesion needs to be treated with endoscopic resection or not. The five reported characteristics of the lesions should be considered on the basis of the following criteria [5].

Size can be measured using biopsy forceps as a reference standard.

The shape can be defined using the Modified Paris Classification, which classifies colonic lesions as polypoid (type 0-I) or non-polypoid (type 0-II) [71].

Polypoid lesions include those which protrude from the mucosa into the lumen ≥2.5 mm. These lesions should be further subgrouped into sessile (0-Is), pedunculated (0-Ip) and semi-pedunculated (0-Isp). Non-polypoid lesions should be defined as slightly elevated (<2.5 mm), completely flat or depressed (referred to as 0-IIa, 0-IIb and 0-IIc, respectively). In particular, non-polypoid lesions greater than 10 mm in diameter are termed “laterally spreading tumors” (LSTs) and require typing based on the appearance of their surface, which may be granular (homogeneous or mixed) or non-granular (flat elevated or pseudo-depressed) [71]. The borders of the lesions should be classified as distinct or indistinct [5].

The surface of the lesions is described by using standardized classifications, such as the Kudo pit pattern and the more recent Frankfurt Advanced Chromoendoscopic IBD lesions (FACILE) classifications [72,73]. The Kudo classification [74] classifies colorectal polyps according to their glandular patterns. Type I pits are described as roundish pits; Type II pits appear as stellar or papillary pits; Type III pits are small roundish, tubular pits, sub-classified into IIIS and IIIL pits based on whether they are smaller or larger than Type I pits, respectively; Type IV pits are described as branch-like or gyrus-like pits; and Type V pits are defined as non-structured pits. Type I and II pit patterns are considered the expression of benign lesions (e.g., normal, hyperplastic or inflammatory polyps), while Type III–V pit patterns are associated with dysplasia occurrence. Modified pit pattern classifications have been proposed in three DCE studies in order to differentiate between neoplastic and non-neoplastic lesions in long-standing IBD [29,75,76], showing high sensitivity and specificity (93–100% and 88–97%, respectively).

The performance of the Japanese magnifying colonoscopy classification (Japan NBI Expert Team (JNET]) for assessing UC-associated neoplasia was evaluated by Kawasaki et al. [77]. Lesions of JNET types IIA, IIB and III were associated with LGD, HGD or superficially submucosal invasive cancer and advanced carcinoma, respectively. When assessing colorectal lesions according to the Kudo classification, pit pattern types III and IV, Vi low irregularity and Vi high irregularity and Vn were associated with LGD/HGD, HGD and advanced colorectal adenocarcinoma, respectively. A more recent Spanish multicenter trial reported that type III–V Kudo pit patterns, as assessed during DCE, are among the predictive factors for CRN (other include sessile morphology, loss of innominate lines and right colon lesions) [78].

The newer validated classification, “FACILE”, includes four characteristics as predictors of neoplastic lesions without using Kudo classification: (1) morphology (nonpolypoid versus polypoid); (2) mucosal surface (roundish, villous and irregular or nonstructural); (3) vessels (nonvisible, regular or irregular or nonstructural) and (4) signs of inflammation [73].

The description of the surroundings includes the evaluation of active mucosal inflammation, other lesions in the surrounding mucosa and features suggesting submucosal fibrosis [5].

The management of dysplastic lesions in colonic IBD is complex, depending on whether the lesion is visible or not, its morphological classification at endoscopy and the histological findings. Potential management options include endoscopic or surgical resection. Overall, visible dysplasia in IBD patients can be managed endoscopically for both polypoid and non-polypoid lesions (although in more select cases) [79], provided the lesion is clearly visible and shows distinct margins [23]. Endoscopic features associated with unresectability include ill-defined margins, features of deep submucosal invasion, asymmetrical lift not related to colitis-associated fibrosis, ulceration or large depressions and flat neoplastic change adjacent to the lesion [56,80].

## 8. Endoscopic Resection Techniques

Endoscopic resection techniques of colorectal lesions, based on their morphology and size, are suggested by ESGE guidelines [81]. Hot snare polypectomy (HSP) is recommended for superficial pedunculated polyps (with or without prophylactic hemostasis, depending on the head size and stalk width). However, cold snare polypectomy (CSP) is recommended for superficial sessile or flat lesions which are <10 mm. In the presence of sessile or flat lesions ≥10 mm without suspicion of submucosal invasion according to the endoscopic appearance of the lesion surface, two endoscopic techniques are recommended; HSP (with or without submucosal injection) is recommended for intermediate lesions (10–19 mm), while endoscopic mucosal resection (EMR), preferably en bloc, is recommended for flat, large lesions (≥20 mm). In case en bloc resection is not feasible or not safe, piecemeal (i.e., excision of the lesion in several stages) EMR can be used. This technique is considered effective and safe, provided that the lesion is removed in as few fragments as possible [82]. Finally, for colorectal lesions with a high suspicion of superficial submucosal invasion, endoscopic submucosal dissection (ESD) should be considered [81]. The specific features of the two endoscopic resection techniques are summarized in Table 1 [81,83].

## 9. Endoscopic Management of Colonic Lesions in IBD

ESGE recommendations have not been tested specifically in an IBD patient setting. In IBD, endoscopic management of visible dysplastic lesions remains challenging regardless of the lesions’ size. The management of visible non-pedunculated dysplastic lesions in these patients is technically more complex than in the general population, particularly in relation to frequently ill-defined margins, scarring and submucosal fibrosis, due to chronic inflammation [84]. These characteristics make endoscopic resection technically demanding and have a higher risk of complications. Inflammatory changes in the colonic mucosa can lead to a difficult demarcation of the target lesion. Moreover, chronic inflammation and the associated fibrosis reduce the submucosal space, making access to the third space during ESD harder. As a result, visualization of the cleavage plane can be compromised, thus increasing the risk of perforation [85].

According to current ECCO guidelines, when possible, dysplastic lesions in an IBD colon should be removed en bloc in order to allow careful histologic assessment [5]. EMR en bloc should be the preferred technique for polypoid or non-polypoid lesions ≤2 cm in diameter, since this technique is easier and associated with a lower rate of complications when compared with ESD. In larger lesions, piecemeal EMR may be necessary. However, in addition to hindering histological evaluation, this may be associated with increased recurrence rates [5]. Therefore, in case of non-polypoid lesions >2 cm, without stigmata of invasive cancer, an en bloc resection through ESD with clear deep and lateral resection margins (i.e., R0) should be performed by expert endoscopists [5]. Overall, current indications for the management of colorectal lesions in IBD have been summarized in Table 2.

These recommendations derive from growing evidence supporting endoscopic resection of visible dysplasia in IBD. Indeed, in the last 10 years, several studies have shown that EMR and ESD for visible dysplastic lesions in IBD patients is both safe and feasible [86,87,88,89,90,91,92,93,94,95,96,97,98]. Although mainly retrospective and including small populations, all of these studies included more than 500 IBD patients in all (mostly with UC), showing about 600 lesions overall. Among these studies, nine focused exclusively on ESD procedures, while four provided information on both ESD and EMR or hybrid ESD (h-ESD) (i.e., partial submucosal dissection followed by snare-assisted resection) procedures. High rates of en bloc resection, complete resection and R0 resection were reported for ESD, with a relatively low rate of complications. These outcomes were also confirmed by a systematic review by Manta et al. [99], considering 216 visible UC-associated dysplastic lesions treated with ESD and reporting an en bloc resection rate of 88.4% (95% confidence interval [CI]: 83.5–92) and R0 resection rate of 78.2% [95% CI: 72.3–83.2].

Moreover, two recent meta-analyses reported a high pooled rate of complete endoscopic resection for dysplastic lesions in IBD patients. Mohapatra et al. [100], considering both EMR and ESD, reported a pooled rate of complete endoscopic resection of 97.9% [95.3–99.7%). A relatively low rate of adverse events (AEs) was reported (pooled rates of 0.8% for endoscopic perforation and 1.6% for bleeding) [100]. These observations suggested that despite the high prevalence of submucosal fibrosis, advanced endoscopic resection is safe and effective in the management of large dysplastic lesions in IBD [100]. Furthermore, local recurrence occurred in approximately 4.9% of patients (higher when the lesions were resected in piecemeal fashion) and was managed by additional endoscopic treatment in most of the cases (94.8%). An additional meta-analysis by Malik et al. [101] aimed to assess the safety and efficacy of ESD only for colorectal dysplasia in IBD patients. The study reported pooled en bloc resection, R0 resection and curative resection rates of 92.5% [87.9–95.45], 81.5% [72.5–88%] and 48.9% [32.1–65.9%], respectively, with a local recurrence rate of 3.9% [2–7.5%]. The pooled rates of bleeding and perforation were 7.7% [4.5–13%] and 5.3% [3.1–8.9%], respectively.

In a recent three-round modified international Delphi consensus, the term colitis-associated neoplasia (CAN) was suggested for “all neoplastic lesions detected in a section of previously or presently inflamed colon” [102]. Neoplastic lesions arising in colonic segments uninvolved with inflammation have been defined as sporadic and appear to not be associated with chronic colitis [103]. Non-polypoid lesions and large (>20 mm) non-pedunculated polyps have been identified as high-risk CAN (HR-CAN) [99]. Indeed, it has been reported that non-polypoid lesions are independent risk factors for advanced neoplasia, while non-pedunculated polyps >20 mm are associated with an increased risk of progression to invasive CRC [104,105,106]. For HR-CAN, the preferred approach is en bloc endoscopic resection, as this technique reduces the risk of recurrence and optimizes histopathological evaluation [102].

Closer endoscopic surveillance is required in this subgroup of IBD patients, particularly those with non-polyploid dysplasia, due to their higher incidence rates of CRC and metachronous dysplasia following endoscopic resection and the risk of local recurrence [107,108].

A study by Chen et al. [107] reported that using a hybrid endoscopic resection technique or ESD for non-polypoid IBD-lesions resulted in pooled en bloc and R0 resection rates of 86% [65–95%] and 70% [55–81%], respectively, with a recurrence rate of 8% [5–13%]. The pooled incidence rates of CRC and metachronous dysplasia were 32.5 [12.2–86.6] and 90.2 [44.9–181.3] per 1000 person-years. Mohan et al. [108] reported a meta-analysis considering 18 studies and a total of 1428 lesions from 1037 IBD patients who underwent endoscopic resection (EMR, ESD or polypectomy). The analysis aimed to assess the pooled risk of incidence and recurrence of neoplastic lesions after resection of colonic dysplasia in patients with IBD. In this meta-analysis, a pooled CRC risk (rate per 1000 person-years of follow-up) of 2 [0–3], a pooled HGD risk of 2 [1–3] and a pooled risk of any lesion of 43 [30–57] were reported. Although CRC and HGD were unusual after endoscopic treatment, the recurrence of any dysplasia was not uncommon, especially for lesions previously treated using EMR. However, many (36.4%) recurrent lesions were managed endoscopically, mostly (55%) using EMR, highlighting the effectiveness of endoscopic resection and subsequent follow-up for IBD patients with dysplastic lesions.

Kaltenbach et al. [109] retrospectively evaluated 326 IBD patients who underwent surveillance endoscopy. The rate of non-polypoid colorectal lesions was 7.7% (63 lesions), the majority of which (96.8% [61 of 63]) were managed endoscopically, with either EMR or ESD, with a low incidence of AEs (1.5%). The authors also provided valuable long-term outcome data, reporting a low rate of recurrence (6.3% [1.8–15.5]), since only four removed lesions were followed by limited local recurrences, which were successfully retreated endoscopically. Moreover, these findings support endoscopic resection and surveillance colonoscopy as being safe and effective for managing non-polypoid colorectal dysplasia in patients with IBD.

Complications during or after endoscopic resection in IBD can be managed as those occurring in the general non-IBD population. Overall, procedure-related bleeding rarely does not allow complete endoscopic resection due to the available techniques and devices for its control. However, perforation can be a more challenging complication of advanced endoscopic resection. Intraprocedural perforation is usually managed endoscopically through clip closure. Depending on the size, shape, depth and site of the mucosal defect, through-the-scope or over-the-scope clips can be used. The treatment should be performed as soon as possible to avoid additional complications. When treated properly, the risk of further complications requiring surgery is indeed reported to be extremely low. Currently, few data are available regarding this issue in IBD, suggesting that endoscopic treatment of intraprocedural perforation does not require further procedures [100,101]. These data are in agreement with those reported in the general non-IBD population. In the two studies specifically addressing this issue, only one case of failure of endoscopic treatment of intraprocedural perforation was in fact reported [110,111]. In cases of delayed perforation after ESD, occurring in up to 2.2% of patients (usually ≤24 h), emergency surgery is required.

## 10. Surveillance After Endoscopic Resection in IBD

Despite strong evidence supporting the need for endoscopic follow-up after endoscopic resection in IBD, no study was aimed directly at comparing different potential surveillance intervals in terms of safety and efficacy. This was also the case in relation to difficulties in matching IBD patients for the main risk factors for new or recurrent colonic lesions (e.g., IBD duration, clinical, endoscopic and histological IBD activity, characteristics of the resected lesion or the above reported general and individual risk factors for CRC).

Consequently, current European recommendations for surveillance are based on retrospective data assessing the recurrence risk of the different lesions [5].

For polypoid lesions and non-polypoid lesions ≤2 cm resected en bloc with EMR, ESD or hybrid ESD, close surveillance with DCE or VCE as well as targeted biopsies is recommended. In case of lesions with findings of HGD upon histological evaluation, a surveillance colonoscopy should be performed every 3 months for the first year and then annually. In case of histologically confirmed LGD, a colonoscopy should be performed every 6 months for the first year and then annually for non-polypoid lesions or directly at 12 months for polypoid lesions <1 cm or pedunculated lesions [5].

Non-polypoid large lesions >2 cm resected with ESD should undergo intense surveillance with DCE or VCE along with targeted and random biopsies every 3–6 months for the first year and then annually [5].

Sporadic adenomas (i.e., polyps which occur in a “non-colitic area”, with both macroscopic and microscopic absence of disease) can also be treated endoscopically, and subsequent surveillance colonoscopy should be performed following post-polypectomy guidelines [5].

## 11. Endoscopic Resection Emerging Techniques

Since 2012, a new “water immersion” EMR technique (underwater EMR (U-EMR)) was developed for removing non-pedunculated colorectal polyps. Technically, after suctioning all of the luminal air, the colonic lumen is filled with water, and then snare resection without submucosal injection is performed [112]. This strategy allows separating the colonic wall layers, thus moving the mucosa and submucosa away from the deep muscular layer while maintaining their normal shape and thickness. A recent meta-analysis suggested that in the general non-IBD population, U-EMR, compared with conventional EMR (C-EMR), is associated with increased rates of en bloc resections with comparable procedural times, recurrence risk and AEs [113]. In IBD patients, U-EMR was reported in case reports to be advantageous for resecting UC-associated neoplasia compared with C-EMR, especially in cases with lesions located in colonic areas characterized by scarring and severe submucosal fibrosis, which can hinder lifting of the lesion [114]. U-EMR showed safety, efficacy and time-saving benefits, effectively removing large polyps in a UC colon with submucosal fibrosis through the “heat-sink” and “floating” effects [115]. Robust data from studies specifically addressing this issue in IBD are lacking. According to findings from the most recent meta-analysis of seven randomized controlled trials, including a total of 1581 polyps, U-EMR in the general non-IBD population was associated with higher rates of en bloc resection (RR 1.18 [1.03–1.35]; I^2^ = 76.6%) when compared with standard techniques [113]. However, these findings need to be confirmed in the IBD population.

Over the last few years, the underwater technique in the general non-IBD population has also been widely used in colorectal ESD (U-ESD) (Figure 1). Despite the still low level of evidence, water immersion has been proven to be as effective and safe as other methods, such as conventional, pocket creation or traction-assisted methods [116]. U-ESD is expected to reduce procedure times, thus providing more effective trimming due to the buoyancy of the lesion. Moreover, U-ESD was also proven to be effective for lesions showing moderate-to-severe submucosal fibrosis [116]. The risks and benefits of U-ESD have still not been tested in IBD. However, in the general non-IBD population, two randomized controlled trials comparing conventional and underwater ESD have been reported [117,118]. Only in one of these studies was a shorter procedural time reported in the U-ESD group [117]. The high frequency of submucosal fibrosis detected in both the conventional and U-ESD groups (67.9% and 60.7%, respectively) [118] may account for the different outcomes between the two studies. Overall, as confirmed in the most recent available meta-analysis comparing outcomes between conventional ESD and U-ESD in the general non-IBD population, U-ESD was suggested to be less time-consuming in the colon-rectum, albeit while showing comparable efficacy and adverse event rates [119]. Considering the high frequency of severe submucosal fibrosis characterizing colorectal dysplastic lesions in IBD, these findings may be even more relevant in this particular population.

Recent evidence suggests that colorectal ESD can be simplified using traction devices [120,121,122], since exposure of the submucosal layer during dissection is one of the main features for a successful ESD, thus improving visibility and allowing safer dissection (reducing the risk of complications, especially perforation) with shorter procedure times. The choice of traction method depends on the size and location of the lesion and the preference and availability of the operator. The use of A-TRACT devices seems promising in IBD-associated lesions [123,124], but high-quality studies aimed at assessing this issue are still not available.

An emerging technique allowing the resection of colorectal lesions not amenable to conventional endoscopic methods is represented by endoscopic full-thickness resection (EFTR). EFTR is realized through the use of a dedicated “full-thickness resection device”, which is a single-use, over-the-scope device which has to be applied to the tip of a standard endoscope. The device consists of a plastic cap preloaded with an over-the-scope clip and an integrated monofilament snare. Different from standard polypectomy, where the snare is used through the working channel of the endoscope, in this device, the snare runs outside the endoscope [125]. EFTR in the colon-rectum is currently mostly used for the treatment of non-lifting adenomas and of adenomatous recurrences. Less common indications include early colonic superficial carcinomas and adenomas localized in colonic areas which are difficult to treat, such as in the appendiceal orifice and in diverticula [125]. Current evidence indicates that the recommended size of lesions prior to resection is ≤30 mm or, in case of severe scarring or inflammation, 20–25 mm [126]. Data regarding the use of this technique in IBD patients are lacking. Baker et al. [127] reported the use of EFTR for removing a 25 mm flat, fibrotic, non-lifting adenoma of the descending colon in a patient with a history of UC. The case describes a successful utilization of EFTR, reporting complete and R0 resection (i.e., uninvolved edges) and no intra- or post-procedural complications (neither bleeding nor perforation).

Endoscopic intermuscular dissection (EID) has been successfully used to treat rectal superficial neoplasia [128]. EID involves dissection of the circular layer from the longitudinal layer of the muscularis propria to achieve a clear vertical dissection margin [129,130]. This technique has recently been reported to lead to high overall technical success and R0 resection rates in sporadic lesions [128], but further experience is required to determine its role for managing CAN.

## 12. Discussion

The development of dysplasia or CRC is one of the most feared complications in patients with colorectal long-standing IBD. Therefore, over the last few decades, dedicated screening programs have been developed in order to achieve early detection of dysplasia, thus leading to timely treatment.

The deeper knowledge of additional risk factors for CRC in patients with IBD has led to the identification of subgroups of patients at higher risk of developing dysplasia and cancer, which therefore deserve more strict surveillance programs. The role of chronically active inflammation in an IBD colon as a relevant risk factor for CRC, even in young patients, has been defined. This observation implies the need for more aggressive medical treatments in this subgroup of patients, aimed at achieving not only clinical but also endoscopic and possibly histological remission of the involved colon. Despite significant improvements in terms of management of IBD activity and endoscopic surveillance, CRC still represents a clinical challenge for physicians, as the mean age at diagnosis is still lower than in the general population.

Surveillance of CRC implies the need for colonoscopy with histological assessment of biopsy samples. During the last few years, technological improvements radically changed endoscopic screening modalities and techniques. The screening protocol indeed shifted from quadrantic biopsies for every 10 cm of an IBD colon to the more recent dye and virtual chromoendoscopy surveillance with target biopsies only for visible lesions. The application of methylene blue or indigo carmine and the adoption of multiple virtual staining provided by endoscope manufacturers (NBI, LCI/BLI and iSCAN), coupled with the introduction of HD endoscopes, allowed better characterization of the colorectal mucosa in IBD patients. Whether dysplasia detection rates differ between dye and virtual chromoendoscopy have been investigated, suggesting substantial overlap between these two techniques. However, differences in terms of time and experience required for performing the procedure and the degree of bowel preparation needed for optimal visualization, together with costs and sustainability, suggest that virtual chromoendoscopy may be more easily and widely used. Moreover, two main issues need to be further investigated, namely the definition of quality indicators for the operators in order to perform adequate surveillance in IBD and the role of quadrantic biopsies. While recommendations exist regarding the requirements for performing optimal VCE in IBD, performance metrics to assess proficiency are still lacking. This is strictly linked to the real need to perform additional random untargeted biopsies. Considering the gain in terms of dysplasia detection allowed by HD technology, DCE and VCE, it is conceivable that random biopsies will be reserved to a minor proportion of selected cases with suboptimal conditions for appropriate surveillance. These currently include colon-related difficulties, sufficient but not excellent bowel preparation, multiple pseudopolyps and, as suggested by SCENIC guidelines, surveillance in patients showing an extremely high-risk of CRC.

Overall, all of these qualitative improvements in terms of available diagnostics tools for early detection of colorectal dysplasia have led to a reduction in the frequency of “invisible dysplasia” in IBD. This allowed the adoption of organ-sparing strategies for the management of superficial colorectal malignancies, including EMR and ESD. While the best strategy for endoscopically managing visible dysplasia in IBD has not been fully defined, it seems clear that all efforts should be driven toward en bloc resection. The advantages of en bloc resection of dysplastic lesions in the general population are well defined. These include lower recurrence rates and, mainly, the feasibility for pathologists to better assess the histology of the lesions (i.e., vertical and lateral margins, leading to a lower risk of missing possible deep infiltration and carcinomatous sites). As in the general population, piecemeal resections should be reserved only for lesions showing an extremely low risk of submucosal invasion. Considering the characteristics of IBD-related lesions and the upsides of en bloc resection, the ER should be planned with this goal. Submucosal fibrosis due to chronic inflammation, vascularization and poor definition of a lesion’s margins may increase the complexity of achieving successful en bloc resection via EMR. Therefore, even current recommendations support limiting the use of EMR to smaller (<20 mm) lesions (Figure 2). However, this size limit may even be reduced, considering the increase in operators’ proficiency and the incomplete resection rate for lesions over 15 mm. Recent initial evidence investigated the possible characteristics predictive of severe submucosal fibrosis in patients with UC, which include a long UC duration and scarring background mucosa [131]. Further identification of the predictive factors for difficult endoscopic resection in IBD patients will allow a proper identification of the more appropriate technique required for endoscopic removal of dysplastic colonic lesions.

Due to the difficulties often encountered during endoscopic resection in IBD, and considering the expertise required to perform advanced resection (EMR and, mainly, ESD) in these patients, these techniques are still not widely adopted. This implies that solid data on post-procedure outcomes, including recurrence of the lesions and complication rates, are limited. Current data regarding the outcome after endoscopic treatment using EMR or ESD suggest a lower risk of incomplete resection and a higher probability of R0 resection using ESD, with expected higher rates of en bloc resection when performing the latter. However, these upsides are balanced by a higher risk related to the procedure, mainly in terms of perforation and, less importantly, bleeding. Perforation risk has been reported to be lower in EMR than in ESD regarding the general non-IBD population. In IBD, the reported perforation risk for EMR indeed ranges from 0.8% to 2.8% and, for ESD, from 3.1% to 8.9% [101]. However, when choosing the best management for IBD-associated superficial lesions, the availability of endoscopic techniques able to manage a potential intraprocedural perforation and the upsides of en bloc resection should also be considered. Local recurrence rates after endoscopic resection (both EMR and ESD) are relatively low (<5%), being potentially treated by further endoscopic treatment in most cases. However, given the relevant risk of metachronous neoplasia, postoperative surveillance colonoscopy is mandatory in these patients.

The clinical implications of the recent shift from surgical to endoscopic resection of visible colorectal dysplasia in IBD are still not defined. Surveillance strategies have changed over the years. Current evidence allowed subgrouping patients according to the risk defined by the presence of IBD-related and patient-related conditions. The better understanding of risk factors for dysplasia and the technological evolution of advanced imaging allowed increasing the surveillance interval in low-risk patients up to 5 years and recommending surveillance colonoscopy every 1, 2 or 3 years, according to the individual characteristics of patients. However, high-quality studies defining the optimal follow-up strategy after ER treatment of dysplastic lesions in IBD are lacking. Current recommendations are based on expert opinions and on the strategies adopted in the general population, suggesting the first follow-up colonoscopy to be performed after 3–6 months and then annually [5,101]. Whether surveillance intervals can be delayed after endoscopic resection of high-risk colitis-associated neoplasia is undefined. When planning surveillance after endoscopic resection, several factors should be considered: the above reported individual risk for CRC, endoscopic and histologic IBD activity over time, endoscopic technique used to remove the lesion and the histology of the resected lesion itself (including R0, R1 or Rx resection).

## 13. Future Directions

During the last few years, a better understanding of the natural history of IBD, including the disease-related risk factors of CRC, led to the optimization of individualized endoscopic surveillance programs in these patients. Accordingly, the better knowledge of the pathogenetic mechanisms, leading to the development of dysplasia and cancer in IBD, allowed significant improvements in terms of medical treatment aimed to reduce both clinical activity and colonic endoscopic and histologic inflammation in these patients. Moreover, due to the growing evidence supporting the effectiveness of endoscopic resection of dysplastic lesions in the general population, such techniques have been further optimized or proposed. The adoption of U-ESD in IBD may be a game changer. This technique may indeed allow better traction and a clearer vision, thus shortening the procedural time and possibly allowing easier dissection, even for lesions with moderate-to-severe submucosal fibrosis. However, the risks and benefits of U-ESD have not been tested for in IBD patients. Regarding techniques allowing better dissection planes, the use of traction devices including A-TRACT appears to be promising in IBD lesions. However, the learning curve required before a wider use of this technique as the real provided advantage, particularly while treating smaller lesions, still needs to be assessed as well. Recently, other devices allowing better traction have been proposed [132], even though they have still not been tested in IBD. Overall, considering that submucosal fibrosis and the achievement of a safe and clear dissection plane are probably the two most relevant issues when performing endoscopic resection in IBD (particularly ESD), the availability of multiple traction devices will be of help in facilitating the procedure. Efforts will be needed in order to assess which of the proposed techniques and devices are most useful in IBD. In cases of difficult lesions with severe fibrosis, the possibility of using the EID technique has been considered only in the rectum. Although IBD data in this regard are still scarce, the reported safety of EID in the general population suggests the potential usefulness of this technique in IBD, as it may overcome some of the difficulties reported when treating IBD-related dysplastic lesions.

## 14. Conclusions

The fast and huge technological and technical evolutions of interventional endoscopy led to recent changes in terms of surveillance strategies in patients with IBD and management of IBD-related dysplastic lesions of the colon. The advent of advanced imaging has lowered the incidental detection of colonic dysplasia over the years, thus allowing the visualization and treatment of previously invisible and therefore unresectable lesions. Indeed, current data are pointing toward a wider spread of organ-sparing strategies for managing dysplasia in IBD. However, more robust evidence is still needed in order to assess the medium- and long-term efficacy of endoscopic resection of visible dysplasia in IBD patients showing a lifelong high risk of local and, more importantly, metachronous recurrence.

## Figures and Tables

**Figure 1 cancers-17-00784-f001:**
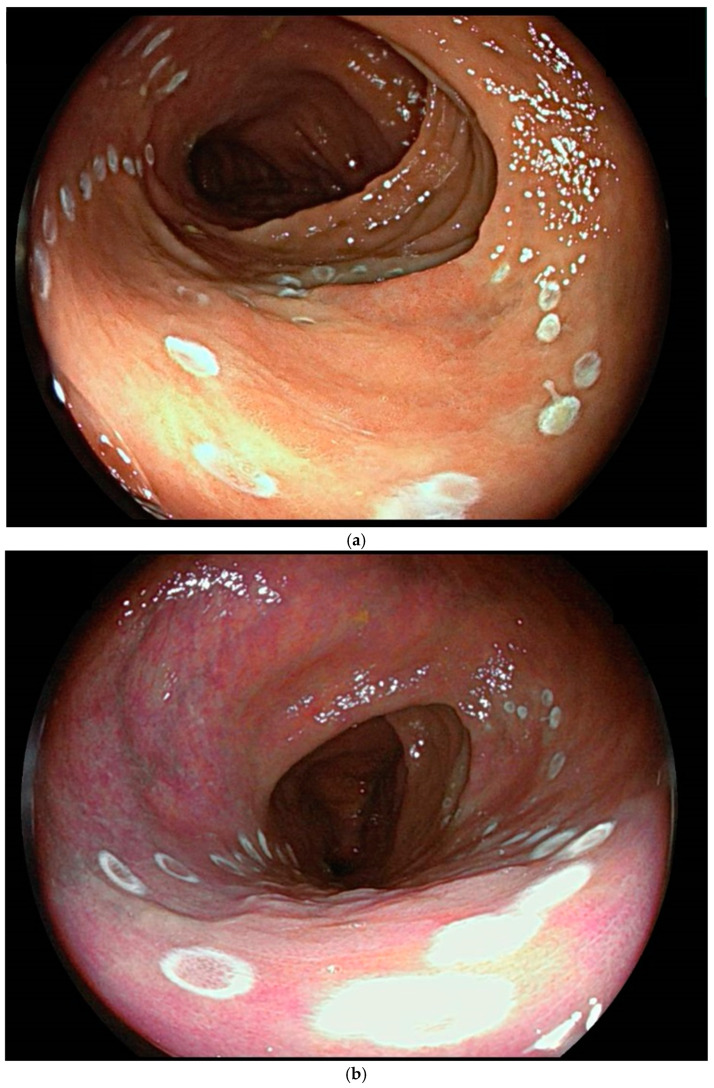
(**a**–**f**) Images of sequential steps of an underwater endoscopic submucosal dissection (U-ESD) of a non-granular flat elevated laterally spreading tumor (LST NG-FE) of the transversus colon in a 68-year-old patient affected by long standing ulcerative colitis. The lesion, involving ¼ of the colonic lumen, extending across 2 colonic folds and presenting a Kudo III pit pattern, was assessed by high-definition white light endoscopy (**a**) and virtual chromoendoscopy (Fujifilm LCI) (**b**). After demarcation (**a**,**b**), the incision of the distal margin of the lesion was performed (**c**), and the submucosa was progressively injected with indigo carmine and saline solution and dissected, showing severe fibrosis (**d**). At the end, the mucosal breach was carefully inspected, and visible vessels were coagulated (**e**). The specimen was then pinned on cork for better histological assessment of vertical and lateral margins (**f**). Histology revealed tubular adenoma with low-grade dysplasia with uninvolved lateral and vertical margins (R0).

**Figure 2 cancers-17-00784-f002:**
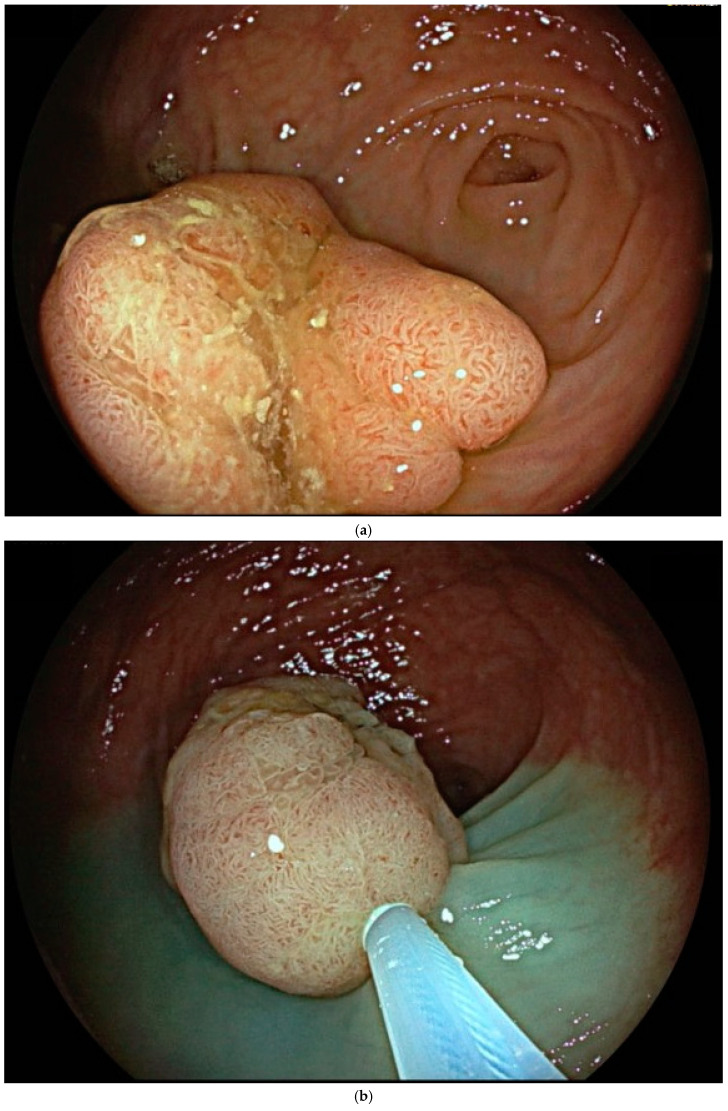
(**a**–**d**) Images of sequential steps of a conventional endoscopic mucosal resection (EMR) of a 15 mm sessile polyp (Paris 0-Is) of the coecum with a type IIIL Kudo pit pattern (**a**) in a 63-year-old female patient affected by long-standing pancolitis, who was in endoscopic remission at the time of the procedure. After submucosal injection of indigo carmine, saline solution and epinephrine (1:250,000), the polyp was resected en bloc (i.e., in one piece) with a multi-fiber snare (**b**). The mucosal breach did not show signs of perforation, and the boarders were free of adenomatous tissue (**c**). At the end, the mucosal defect was completely closed endoscopically with 4 clips to prevent delayed bleeding (**d**). Histology revealed tubulo-villous adenoma with low-grade dysplasia, with dysplasia-free lateral and vertical margins (R0).

**Table 1 cancers-17-00784-t001:** Endoscopic mucosal resection (EMR) and endoscopic submucosal dissection (ESD): main technical features.

Endoscopic mucosal resection (EMR) involves injection of a solution into the submucosal space in order to separate the mucosal lesion from the underlying muscularis propria. The lesion can then be resected by snare electrosurgery. The submucosal cushion theoretically reduces the risk of thermal or mechanical injury of the underlying muscularis propria [81].	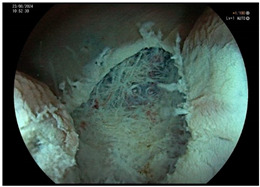
Endoscopic submucosal dissection (ESD) is characterized by three steps: fluid injection into the submucosa in order to swell the lesion from the muscle layer and circumferential cutting of the mucosa surrounding the lesion, followed by dissection of the submucosal connective tissue beneath the lesion (standard technique). The major advantages of this technique in comparison with polypectomy or EMR include control of the resected size and shape, possible en bloc resection even in a large lesion and also resectability of the lesions in case of submucosal fibrosis. Disadvantages include the time-consuming procedure associated with a higher risk of complications (mainly bleeding and perforation) when compared with EMR [83].	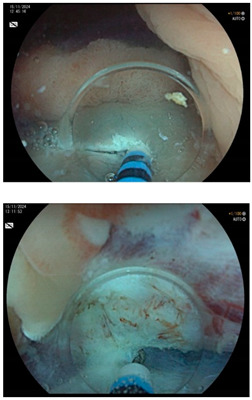

**Table 2 cancers-17-00784-t002:** Summary of currently recommended therapeutic management of visible dysplastic IBD-associated lesions.

Lesion’s Appearance	Suggested Treatment
Simple (e.g., pedunculated) and small (<10 mm) polypoid lesions	Standard polypectomy techniques (CSP, HSP)
Polypoid and non-polypoid lesions ≤20 mm without stigmata of invasive cancer or distinctive borders	En bloc EMR (preferred) ESD or hybrid ESD (in case of poor lifting, high risk of non-en bloc resection by EMR)
Non-polypoid lesions >20 mm without stigmata of invasive cancer or distinctive borders	ESD Surgery (in patients at higher risk, difficult colorectal sites, ESD failure, no available referral expert centers)
Lesion with indistinctive borders or suggestive of submucosal invasive cancer	Surgery

Abbreviations: CSP = cold snare polypectomy; HSP = hot snare polypectomy; EMR = endoscopic mucosal resection; ESD = endoscopic submucosal dissection.

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
