# Peer review of "Colitis-Associated Dysplasia in Inflammatory Bowel Disease: Features and Endoscopic Management"

_cancers, 2025, doi:10.3390/cancers17050784_

Round 1

Reviewer 1 Report

Comments and Suggestions for Authors

thank you for allowing me to review this update on the management of pre-cancerous colorectal lesions in chronic inflammatory bowel disease. this comprehensive update is well written and complete. however, it would be easier to read if it were illustrated with tables and decision algorithms, including the place of surgical treatment.

Comments on the Quality of English Language

no comment

Author Response

Reviewer 1.

Thank you for allowing me to review this update on the management of pre-cancerous colorectal lesions in chronic inflammatory bowel disease. This comprehensive update is well written and complete. However, it would be easier to read if it were illustrated with tables and decision algorithms, including the place of surgical treatment.

We wish to thank reviewer 1 for considering the submitted review comprehensive, updated, well written and complete. As requested, in the revised version of the manuscript, decision algorithms, including the place of surgical treatment, are summarized in a new Table 2.

Reviewer 2 Report

Comments and Suggestions for Authors

This review seems to support the idea that improvements in endoscopic methods allow adequate assessment and management of dysplasia and cancer in IBD subjects to be used routinely.  The progress in methods is laid out with abundant detail from clinical trials.  A couple of points concerned me.  It is suggested but not made crystal clear that random sampling is no longer needed with the newer available methods.  Is this true?  The upgrades detect what was once skipped over?  The percentage of perforation with ESD seems very high.  My own search for this confirmed it, 9% upper limit with 2.2% delayed.  How big a risk is this for the patient once it happens? 

Author Response

Reviewer 2.

This review seems to support the idea that improvements in endoscopic methods allow adequate assessment and management of dysplasia and cancer in IBD subjects to be used routinely. The progress in methods is laid out with abundant detail from clinical trials. A couple of points concerned me.

  1. a) It is suggested but not made crystal clear that random sampling is no longer needed with the newer available methods. Is this true? The upgrades detect what was once skipped over?
  2. b) The percentage of perforation with ESD seems very high. My own search for this confirmed it, 9% upper limit with 2.2% delayed. How big a risk is this for the patient once it happens?

 We wish to thank the reviewer for giving us the opportunity to further address this issue.

  1. a) As required, in the revised version of the manuscript, the role of random sampling has been further clarified, as follows (revised pages 8-9): “Differently, random sampling of the colon-rectum is no longer required. Indeed, data from two RCTs and a retrospective cohort study reported the same dysplasia detection rate in patients undergoing VCE when comparing random plus targeted biopsies versus target biopsies alone [38-40]. The diagnostic yield of dysplasia detection rate by using random biopsies was reported to be as low as 0.2% [41]. In very high-risk patients (i.e. previous dysplasia, concomitant PSC, active inflammation, scarred colon) quadrantic biopsies every 10-cm are still indicated [5]. In this particular sub-population, the addition of random biopsies is associated with an increased diagnostic yield of colorectal dysplasia [41]. Other from very high-risk patients, random biopsies in addition to VCE or DCE should be reserved only for special situations, such as surveillance after endoscopic resection of large (>2 cm) non-polypoid large lesion [5].”
  2. b) In the revised manuscript, additional data requested by the reviewer were added, as follows (page 19-20): “Complications during or after endoscopic resection in IBD can be managed as those occurring in the general non-IBD population. Overall, procedure-related bleeding rarely does not allow a complete endoscopic resection, due to available techniques and devices for its control. Differently, perforation can be a more challenging complication of advanced endoscopic resection. Intraprocedural perforation is usually managed endoscopically by clip closure. Depending on the size, shape, depth and site of the mucosal defect, through-the-scope or over-the-scope clips can be used. The treatment should be performed as soon as possible to avoid additional complications. When treated properly, the risk of further complications requiring surgery is indeed reported as very low. Currently, few data are available regarding this issue in IBD, suggesting that endoscopic treatment of intraprocedural perforation does not require further procedures [101,102]. These data are in agreement with those reported in the general non-IBD population. In the 2 studies specifically addressing this issue, only 1 case of failure of endoscopic treatment of intraprocedural perforation was indeed reported [111,112]. In case of delayed perforation after ESD, occurring in up to 2.2% of patients (usually ≤24 hours), emergency surgery is required.” As requested, in the revised text it has also been further specified as follows (page 26): “Perforation risk has been reported to be lower in EMR than in ESD, as for the general non-IBD population. In IBD, the reported perforation risk for EMR indeed ranges from 0.8% to 2.8% and for ESD from 3.1% to 8.9% [102]. However, when choosing the best management for IBD-associated superficial lesions, the availability of endoscopic techniques able to manage a potential intraprocedural perforation and the upsides of en-bloc resection should also be considered”.

Reviewer 3 Report

Comments and Suggestions for Authors

The manuscript provides a thorough and well-structured review of colitis-associated dysplasia in inflammatory bowel disease, covering epidemiology, carcinogenesis, surveillance strategies, and endoscopic management techniques. The extensive literature review strengthens the discussion, though some sections could benefit from more concise synthesis to enhance readability. The discussion on endoscopic resection techniques is comprehensive, but further clarification on decision-making criteria between EMR and ESD in specific clinical scenarios would be valuable. Additionally, while novel techniques such as U-EMR and traction-assisted ESD are well described, their practical implementation in clinical settings and comparative efficacy with conventional methods could be more explicitly discussed. Ensuring uniform terminology, particularly in distinguishing sporadic from colitis-associated neoplasia, would improve clarity. Overall, the manuscript is highly informative, but minor refinements in structure and clarity could further enhance its impact

Author Response

Reviewer 3.

The manuscript provides a thorough and well-structured review of colitis-associated dysplasia in inflammatory bowel disease, covering epidemiology, carcinogenesis, surveillance strategies, and endoscopic management techniques.

  1. a) The extensive literature review strengthens the discussion, though some sections could benefit from more concise synthesis to enhance readability.
  2. b) The discussion on endoscopic resection techniques is comprehensive, but further clarification on decision-making criteria between EMR and ESD in specific clinical scenarios would be valuable.
  3. c) Additionally, while novel techniques such as U-EMR and traction-assisted ESD are well described, their practical implementation in clinical settings and comparative efficacy with conventional methods could be more explicitly discussed.
  4. d) Ensuring uniform terminology, particularly in distinguishing sporadic from colitis-associated neoplasia, would improve clarity.

Overall, the manuscript is highly informative, but minor refinements in structure and clarity could further enhance its impact

We wish to thank the reviewer for considering the submitted manuscript as providing a thorough and well-structured review of colitis-associated dysplasia in inflammatory bowel disease, covering epidemiology, carcinogenesis, surveillance strategies, and endoscopic management techniques. As requested:

  1. a) The text has been reviewed and shortened in order to allow a better readability.
  2. b) As also requested by reviewer 1, a new Table 2, summarizing the indications for the treatment of colitis-associated dysplasia in IBD, has been added to further clarify decision-making criteria between EMR and ESD in specific clinical scenarios.
  3. c) We do agree with the reviewer that more insights regarding the implementation in clinical practice of U-EMR and traction assisted ESD compared to standard techniques may be very useful. However, by our knowledge, these data in IBD are still not defined. Therefore, currently available evidences regarding the characteristics of IBD-associated colorectal lesions have been reported in the revised version of the manuscript as follows (Page 21-22): “Robust data from studies specifically addressing this issue in IBD are lacking. Findings from the most recent meta-analysis of 7 randomized controlled trials, including a total of 1581 polyps, U-EMR in the general non-IBD population was associated with higher rates of en-bloc resection (RR 1.18 [1.03-1.35]; I²= 76.6%) when compared to standard technique [114]. However, these findings need to be confirmed in the IBD population.” In the revised version it has also been added as follows (Page 22): “Risks and benefits of U-ESD have still not been tested in IBD. However, in the general non-IBD population, 2 randomized controlled trials comparing conventional and underwater ESD have been reported [118, 119]. Only in one of these studies a shorter procedural time was reported in the U-ESD group [118]. The high frequency of submucosal fibrosis detected in both conventional and U-ESD group (67.9% and 60.7%, respectively) [119] may account for the different outcome between the 2 studies. Overall, as confirmed in the most recent available meta-analysis comparing outcomes between conventional ESD and U-ESD in the general non-IBD population, U-ESD was suggested to be less time consuming in the colon-rectum, although showing a comparable efficacy and adverse events rates [120]. Considering the high frequency of severe submucosal fibrosis characterizing colorectal dysplastic lesions in IBD, these findings may be even more relevant in this particular population.”
  1. d) We thank the reviewer for this helpful comment. In the revised version, efforts have been made in order to avoid confusion between sporadic from colitis-associated neoplasia. Overall, findings from the present review refer to IBD-associated dysplasia unless differently specified.